# Adaptation of a Model Spike Aptamer for Isothermal Amplification-Based Sensing

**DOI:** 10.3390/s24216875

**Published:** 2024-10-26

**Authors:** Emre Yurdusev, Pierre-Luc Trahan, Jonathan Perreault

**Affiliations:** Armand-Frappier Santé Biotechnologie Research Centre, INRS (Institut National de la Recherche Scientifique), Laval, QC H7V 1B7, Canada; emre.yurdusev@inrs.ca (E.Y.); pierre-luc.trahan@inrs.ca (P.-L.T.)

**Keywords:** aptamer, isothermal amplification, hyperbranched rolling circle amplification (HRCA), loop-mediated isothermal amplification (LAMP), SARS-CoV-2, Spike protein, colorimetric detection, point-of-care test, allostery, structure dynamics

## Abstract

Isothermal amplification (IA) techniques like rolling circle amplification (RCA) and loop-mediated isothermal amplification (LAMP) have gained significant attention in recent years due to their ability to rapidly amplify DNA or RNA targets at a constant temperature without the need for complex thermal cycling equipment. Such technologies, combined with colorimetric systems rendering visual confirmation of the amplification event, are ideal for the development of point-of-need detection methods suitable for field settings where access to specialized laboratory equipment is limited. The utility of these technologies, thus far limited to DNA and RNA targets, could be broadened to a wide range of targets by using aptamers. Composed of DNA or RNA themselves, aptamers can bind to substances, including proteins, metabolites, and inorganic substances. Their nucleic acid nature can potentially allow them to serve as a bridge, extending the reach of DNA/RNA-centric technologies to the broader molecular world. Indeed, the change in aptamer conformation occurring during ligand interaction can be used to elaborate ligand-responding RCA or LAMP templates. By using an existing aptamer targeting SARS-CoV-2 Spike protein as a model, we explored the possibility of establishing ligand-responsive IA systems. Our study used aptamers with simple sequence modifications as templates in LAMP assays and hyperbranched RCA (HRCA) by exploiting the dynamic nature of the model aptamer to trigger these IA systems. Importantly, our work uniquely demonstrates that this aptamer’s dynamic response to ligand binding can regulate both RCA and LAMP processes. This novel approach of using aptamer conformational changes to trigger LAMP paves the way for new aptamer-based detection assays. Our system detects 50 nM of Spike protein, with LAMP occurring within 30 min in the presence of Spike. The colorimetric readout showed clear results, allowing for the detection of Spike protein presence.

## 1. Introduction

Isothermal amplification (IA) techniques, such as rolling circle amplification (RCA) or loop-mediated isothermal amplification (LAMP), offer a powerful means of amplifying specific DNA targets (or RNA with the incorporation of an additional reverse transcription step for LAMP [1,2]), functioning effectively at a constant temperature [3,4,5,6,7]. Such methods provide a level of sensitivity akin to PCR but require less equipment [1,2,8,9]. There exists a variety of simplified, low-power solutions capable of maintaining the thermal conditions necessary for IA, allowing for field applications [1,2,8,9] or even IA at ambient temperatures, which further simplifies the process [10,11]. The development of new detection or diagnostic systems based on colorimetry, such as pH-based systems [12,13,14,15], which provide a visual confirmation of ongoing DNA amplification, is making substantial strides for portable point-of-use assays [1,16,17,18]. When combined with IA, these systems enable faster and simpler detection of targets, enhancing the practicality and utility of these approaches [1,16,17].

The distinctive attributes of aptamers present an exceptional opportunity to expand upon these technological developments. The traditional sphere of IA has been confined to DNA or RNA targets. However, due to their nucleic acid structure, aptamers are compatible with IA systems, allowing them to act as effective triggering elements. Moreover, aptamers can interact with a wide variety of target molecules beyond nucleic acids [19,20,21,22]. This dual capability enables the construction of IA systems that are governed by the interaction of aptamers with their ligands. Consequently, IA can be triggered by the presence of diverse molecular targets, extending the detection spectrum beyond nucleic acids and enabling the design of innovative IA systems responsive to a broader range of chemical compositions.

Examples of this concept have been demonstrated, primarily with RCA, for the detection of a variety of targets including thrombin [23] and histone-modifying enzymes [24], as well as for the ultra-sensitive detection of cardiac Troponin I [25], among others. Here, aptamers were used as static elements, so their capacity to initiate RCA systems remains unaffected by any potential interactions with ligands. These examples utilize aptamers as static elements, and they are effective in triggering RCA. However, these approaches require several additional experimental steps. For example, studies like those by Li et al. [25]. and Jiang et al. [24] require multiple washing steps (e.g., three washes per cycle in Li’s study) to remove unbound elements, which adds 15–30 min per cycle. In addition, blocking steps to prevent non-specific binding, such as the 1 h blocking at 4 °C in Li et al.’s study [25] or the 2 h mercaptohexanol incubation in Di Giusto et al.’s study [23], further extend assay time. Moreover, extra incubation steps for secondary or tertiary elements, like the overnight RCA incubation in Li et al.’s study [25], significantly lengthen the procedure. In total, such static aptamer systems can add 3–5 h to the workflow, making them less efficient for rapid diagnostics.

Other strategies capitalize on the dynamic conformational change in aptamers when interacting with a ligand to initiate RCA processes. Here, RCA only occurs in response to the interaction of aptamers with their specific target, thus eliminating the need for washing steps. For instance, Yang et al. [26] redesigned the Platelet-Derived Growth Factor (PDGF) aptamer to undergo a conformational change into a circular structure in the presence of PDGF. Similarly, Wu et al. [27] utilized an aptamer that binds to both ends of a target oligonucleotide through base-pair interactions. In the absence of ATP, the aptamer’s structure allows the 3′ and 5′ ends of the oligonucleotide to remain unconstrained and not align head-to-head. Upon binding to ATP, the aptamer undergoes a conformational change to a more compact structure, pulling the 3′ and 5′ ends into a head-to-head position. This alignment facilitates the ligation process, which then initiates rolling circle amplification (RCA), streamlining the process and reducing time significantly compared to static approaches. Bialy et al. [28] employed a strategy using an aptamer as a primer for RCA, which can only prime RCA when it is not interacting with its protein ligand. These methodologies harness RCA for versatile use of aptamers, offering faster and simpler procedures compared to traditional static systems.

Surprisingly, even though LAMP is the most commonly used IA assay and presents some advantages over RCA, such as faster reaction times and higher amplification efficiency [29,30], as well as the fact that it does not require ligation, studies using aptamer-based LAMP systems are limited. LAMP can amplify DNA with high yield within 30–60 min compared to the longer and more linear process of RCA. Even in the case of hyperbranched rolling circle amplification (HRCA), which increases amplification yield through branching, LAMP remains faster due to its use of multiple primers forming loop structures that drive rapid cycling and amplify DNA more efficiently [30]. These attributes make LAMP a more suitable option for time-sensitive applications where both speed and efficiency are critical. Zhang et al. [31] used an aptamer-based system to detect the presence of thrombin, triggering a cascade of enzymatic events and generating the primers necessary for the LAMP process. Otherwise, the combination of LAMP systems with aptamer technology has not been widely studied, unlike RCA. However, the approach used by Zhang et al. relied on several enzymatic steps and used the aptamer in an indirect way (via a primer generation principle) to perform LAMP, making it relatively complex to implement.

In this study, we further expand upon these advances by using a model aptamer binding SARS-CoV-2 Spike glycoprotein (Spike aptamer) [32] as a template, triggering both LAMP and RCA systems in a ligand-dependent manner, using a simpler and faster system than the previous IA aptamer-triggered assays [26,27,31]. Furthermore, employing a colorimetric system, we visually confirmed the presence of the Spike protein in the test samples at concentrations within the nanomolar range using an adapted design derived from this model aptamer.

## 2. Materials and Methods

### 2.1. Structure Prediction

For secondary structure predictions, we employed the UNAFold Web Server [33], tailoring the software settings to simulate the distinct conditions pertinent to RCA and LAMP procedures. At 20 °C, which is the typical initiation temperature for RCA, we adjusted the sodium ion (Na^+^) concentration to 73.5 mM and the magnesium ion (Mg^2^⁺) concentration to 5 mM. This reaction occurred in a mixture involving the aptamer-binding buffer diluted with T4 DNA ligase buffer from New England Biolabs (NEB, catalog number B0202, Ipswich, MA, USA). In contrast, for the 60 °C predictions, aligning with the LAMP procedure, we maintained the Na^+^ concentration at 73.5 mM but increased the Mg^2^⁺ concentration to 8.3 mM, with the reaction occurring in a blend of the aptamer-binding buffer and the WarmStart^®^ Colorimetric LAMP 2X Master Mix (NEB, catalog # M1804).

### 2.2. HRCA and LAMP Assay Procedures

For our experiments, both HRCA and LAMP assays were performed using aptamers or corresponding control oligonucleotides (with sequences detailed in Appendix A) at a concentration of 5 pM. We employed the Spike protein (Creative BioMart, catalog # Spike-145V, New York, NY, USA) and control proteins (Cellulase B, kindly provided by Professor Nicolas Doucet’s laboratory) at a concentration of 50 nM in the assay mixtures. The primers used in these assays, detailed in Appendix A, were added at a concentration of 5 µM.

Prior to conducting the HRCA and LAMP tests, both aptamers and control oligonucleotides underwent a heat shock cooling cycle. This procedure, recommended by Song et al. [32] for their aptamer (from which our EY615 aptamer design was derived), involved heating the nucleotides in PBS buffer with 0.55 mM MgCl_2_ at 95 °C for 5 min, before rapidly cooling to 4 °C. The nucleotides were then incubated at room temperature for 20 min in the same buffer, either with or without the addition of Spike protein or Cellulase B protein.

For the HRCA assays, we used T4 DNA ligase (NEB) and a custom-made T4 DNA ligase buffer without Tris-HCl to avoid interference with colorimetric tests. The ligation process consisted of a 10 min incubation at room temperature. Either immediately following this step for the RCA or directly after the Spike protein incubation for the LAMP assays, the primer oligonucleotides and the WarmStart^®^ Colorimetric LAMP 2X Master Mix (NEB, catalog no: M1804) were introduced into the mixture. The assays were then incubated for 30 min at 60 °C before the reactions were halted by transferring the assays to an ice bath.

### 2.3. Colorimetric Analysis in HRCA and LAMP Assays

For the colorimetric evaluation of our HRCA and LAMP assays, we allocated 10 µL of each assay into a 1536-well plate, specifically using the Greiner Flat Bottom Black Polystyrene plate (Millipore Sigma, Burlington, MA, USA, Cat. No.: 782096/782097). The colorimetric data for each sample were acquired utilizing the Tecan Infinite^®^ M1000 plate reader (Tecan, Männedorf, Switzerland). This reader was set to perform absorbance scans in the range of 380 nm to 800 nm, employing a wavelength increment of 5 nm, resulting in 85 reads for each sample. We configured the device with 25 flashes per reading and a settle time of 1 ms, and allowed the rest of the parameters to be adjusted through the plate reader’s auto-calibration feature.

To quantify the color change in our assays, which use a red phenol dye-based colorimetric buffer, we measured optical absorption (Abs) at two key wavelengths: 560 nm for red/purple hues (Abs_560_) and 430 nm for yellow hues (Abs_430_). By calculating the ratio of Abs_430_ to Abs_560_, we were able to numerically represent the color transition from red/purple to yellow, which is crucial for assessing the outcomes of our colorimetric assays.

### 2.4. Fluorescence Measurement Techniques in Mechanistic Studies

In our mechanistic studies, we used the EY615QF aptamer, specifically tagged at the 3′ end with a Black Hole Quencher^®^-2 (BHQ2) molecule and at the 5′ end with Cyanine5™ (Cy5) from Invitrogen (Waltham, MA, USA), supplied by IDT DNA Technology, Coralville, IA, USA. The fluorescence readings were conducted using the GE Typhoon FLA 9500 scanner from GE Healthcare (Chicago, IL, USA). This scanner was set to an excitation wavelength of 635 nm, coupled with LPR emission filters for optimal detection.

Fluorescence quenching and de-quenching (EY615QF + 10X complement EY620) tests were conducted following a heat shock and cool down process (95 °C for 5 min, followed by 5 min at 4 °C). EY616QF aptamer (6.5 nM when it is not specified otherwise) with and without Spike protein (100 nM) was diluted in PBS with 0.55 mM MgCl_2_ (as per Song et al.’s [32] recommendation for aptamer–Spike interaction). RCA/LAMP primers (EY616 and EY617) were used, with a quantity of 3 µM each (sequence details in Appendix A).

For the quantification of the fluorescence signals, we employed densitometry techniques using the ImageQuant TL version 8.1 software, also from GE Healthcare. To ensure the accuracy of our fluorescence readings, we first determined the background signal using wells that contained only the assay buffer. This background signal was then subtracted from the fluorescence values of the assay samples to facilitate a more precise and meaningful comparison of the data.

## 3. Results

### 3.1. Spike Aptamer as a Template for RCA and LAMP Systems: Structural Compatibility and the Modulating Influence of Spike

#### 3.1.1. Aptamer Design Rational

Inspired by the work of Song et al. [32] who proposed one of the first aptamers targeting the Spike protein of SARS-CoV-2, we designed minor sequence modifications in order to use them for IA. These designs were inspired specifically by one of these original structures [32]: a derivative of the full-length aptamer truncated in the regions of the SELEX primers’ binding sites. This aptamer had the desired structural features suitable for RCA-based amplification. This truncated variant exhibited a predicted dumbbell-like secondary structure at room temperature (similar to the configuration represented in Figure 1A marked with *), characterized by the nearly head-to-head orientation of its terminal nucleotides, with an extra nucleotide lingering on the 3′ end. We trimmed this extra nucleotide to create our own model aptamer design, EY615, exhibiting a predicted optimal head-to-head alignment of extremities compatible with ligation for RCA. Additionally, we developed another design, EY615-5′C, by removing the C nucleotide at the 5′ extremity. This alteration was made because the reverse complementary sequence of EY615-5′C presents structures more compatible with a LAMP-like amplification, as illustrated in Appendix A. The presence of the 5′C nucleotide in EY615, which becomes 3′G in the reverse complementary sequence, was found to render some structure predictions incompatible with the LAMP scheme because it is not base-paired and therefore does not act as a good priming site.

#### 3.1.2. EY615 Can Serve as Template for LAMP and HRCA

Our results, first and foremost, demonstrated that our aptamer designs, namely EY615 and EY615-5′C, performed effectively as templates for both HRCA and LAMP assays. As monitored by pH-based colorimetric assay methods [12,13,15], these designs drove significant color changes (Figure 2 and Appendix A for EY615-5′C-related results), akin to those observed with positive controls (EY615rdl, a version of EY615 with reinforced disposition to form a “dumbbell-like” structure templating both HRCA and LAMP; see Appendix A). In contrast, nucleotide sequences lacking the properties conducive for signal amplification (e.g., EY615lv, the full-length aptamer form [32] where 3′ and 5′ extremities are not in a “head-to-head” configuration) did not induce any significant color transition (Figure 2D).

In the context of both HRCA and LAMP systems, we quantified the color switch by comparing optical absorbance (Abs) measurements at wavelengths revealing the unamplified (red/purple state at wavelength 560 nm, Abs_560_) and amplified states of the assay (yellow state at wavelength 430 nm, Abs_430_) [15,34] (Figure 2A). The ratio of red/purple to yellow absorbance was used as a measure of the amplification with higher ratios, indicating elevated amplification. Relative to the corresponding negative controls (EY615lv), we observed significant increases in this Abs_430_/Abs_560_ ratio in both LAMP and HRCA contexts. In the LAMP context, EY615 showed a 69% increase in the ratio, while EY615-5′C demonstrated a 62% increase. In the HRCA context, EY615 exhibited a more substantial 139% increase in the ratio. These percentages represent the maximum increases in the Abs_430_/Abs_560_ ratio observed in assays where the color switch was most pronounced (Figure 2B and Appendix A for detailed raw data).

To further validate these colorimetric observations, we conducted agarose gel electrophoresis of the amplification products (Figure 2C). In both LAMP and HRCA assays, tests performed with EY615 that exhibited a color switch also demonstrated corresponding gel electrophoresis profiles indicative of ongoing amplification. Conversely, assays where no color switch was observed showed minimal or no bands in the gel analysis.

#### 3.1.3. Spike Dependence of LAMP and HRCA

The very minor modification (e.g., deleting the 3′ single unpaired base) that we made to the previously published Spike aptamer [32] was not expected to affect Spike binding. Indeed, our study reveals the influence of Spike on both HRCA and LAMP, confirming that we preserved the interaction between Spike and our aptamer design. Spike was found to enhance LAMP (Figure 2), substantiated by both agarose gel electrophoresis and colorimetric measurements. Specifically, the Abs_430_/Abs_560_ color switch ratio increased by 81% for LAMP in the presence of Spike, compared to assays without Spike, for the EY615 tests (Figure 2 and Appendix A for detailed raw data), correlating with the agarose gel analysis and the observed bands.

In the HRCA tests, our observations indicated that the presence of Spike inhibits HRCA amplification, opposite to its effect on LAMP. However, this inhibitory effect was partially obscured by LAMP occurring in the background. As a result, we observed color switches and gel profiles that did not match what would be expected in an isolated Spike-inhibited HRCA scenario. The ladder-like pattern observed in the HRCA assays, which is sometimes seen as a result of HRCA [35,36], is more likely due to a background LAMP event, as this multi-banded pattern is highly characteristic of LAMP. This is not surprising since the HRCA procedure is similar to LAMP in many ways, with the only difference being the additional ligation step for HRCA, which does not inhibit LAMP from occurring. LAMP can occur in several ways in the HRCA process, as represented in Figure 2E.

If HRCA is inhibited in presence of Spike, two scenarios are possible: (i) with LAMP-compatible primers (LCPs), LAMP can still occur (assays 1 and 2, Figure 2); (ii) with HRCA-specific primers (HSPs, Appendix A for primer design details), LAMP does not occur (assays 5 and 6, Figure 2).

Conversely, in the absence of Spike, when HRCA is triggered, even if LAMP is initially inhibited, the accumulation of HRCA products eventually facilitates LAMP. In assays with LCP (assays 3 and 4, Figure 2), the initial inhibitory effect on LAMP is counterbalanced by the high accumulation of HRCA products, which serve as templates for LAMP. In assays with HSPs (assays 7 and 8, Figure 2), which are not supposed to support LAMP, we hypothesize that LAMP-triggering templates are produced, especially considering the inherent potential of HRCA products to initiate LAMP. When displaced by the polymerase, the 3′ branched RCA product can potentially fold on itself to self-prime, allowing for LAMP-like amplification (see EY615 complementary reverse strand structure prediction in Figure 1 and Appendix A).

Consequently, the inhibitory effect of Spike on HRCA becomes evident and quantifiable only when using HRCA-specific primers (HSPs), which are incompatible with LAMP. In assays with HSPs, we observed a 41% decrease in the Abs_430_/Abs_560_ ratio in the presence of Spike compared to assays without Spike (Figure 2, Appendix A). This significant reduction clearly demonstrates Spike’s negative impact on HRCA amplification.

To investigate the specificity of the observed Spike effects, we performed control LAMP tests with two other proteins, cellulase B (CellB) and p53, to determine if the Spike effect on the LAMP process could be replicated by other proteins. The results showed that this is not the case, as evidenced by the lack of colorimetric changes (Figure 2D). We also investigated whether Spike’s effects were specific to its interaction with the aptamer EY615. To test this, we used a scrambled aptamer sequence (EY615scr) still designed to be LAMP-compatible (Appendix A) but without the Spike-binding motif [32]. The results with EY615scr (Appendix A) showed slight and variable amplification profiles that were not correlated with the presence of Spike, reinforcing the specificity of the Spike–EY615 interaction.

Additionally, we used a longer version of the aptamer, unable to form the necessary dumbbell-like structure for HRCA and LAMP triggering, as a negative control (EY615lv), and also used a positive control (EY615rdl), which has a reinforced dumbbell-like structure forming sequence (Appendix A for sequence and structure details). The negative control showed no amplification, while the positive control exhibited strong amplification, as expected (Figure 2D).

We tried to improve the Spike LAMP assay by deleting the 5′C (EY615-5′C), which leads to an unpaired 3′C in the most highly predicted structures of the EY615-complementary DNA sequence and thus would likely inhibit continuation of LAMP cycles. However, we observed a reduced difference in LAMP between samples with and without Spike (Appendix A). Moreover, instead of the expected increase in amplification in the “+Spike” samples, we noted an increased amplification rate in the “-Spike” samples. So, while the EY615-5′C aptamer has an enhanced ability to serve as a LAMP template, it is at the cost of no longer requiring Spike.

Our findings suggest that Spike acts more as a facilitator for the LAMP process rather than being an essential component for its initiation. It appears EY615 is not optimal for standard LAMP, thus making the contribution of Spike necessary, which would explain the improved readings obtained with EY615 compared to EY615-5′C, which depends less on Spike for the LAMP initiation.

### 3.2. Mechanism of EY615-Spike Interaction and Its Impact on LAMP and RCA

#### 3.2.1. Experimental Rational and Setup

To uncover how Spike interacts with EY615, and the ensuing impact on LAMP and RCA, we utilized our EY615 aptamer design tagged with Cy5 at the 5′ end and a BHQ2 quencher on the 3′ end (namely EY615QF). This design allowed us to monitor changes in the aptamer structure through the measurement of fluorescence. In this setup, a head-to-head (or dumbbell-like, as in the structure predictions depicted in Figure 1A, * and **) aptamer configuration quenches Cy5 fluorescence, whereas a different structure, such as an open configuration (Figure 1A, ***), triggers a fluorescence signal.

Our first experiment was designed to verify the self-quenching of EY615QF. When EY615QF is exposed to a complementary sequence (EY620, see Appendix A), compelling it to adopt an open structure, we observed a notable increase in fluorescence (Appendix A). These tests were conducted in the same buffer environment that was used for the previous Spike interaction with RCA and LAMP.

#### 3.2.2. Spike Effect on Aptamer Configuration

For the assays involving Spike, we used 6.5 nM EY615QF to ensure Spike (100 nM) was present in excess. Under these conditions, the fluorescence shifts we observed were in a comparable range to those seen within the concentration range of 1–10 nM for the aptamer EY615QF in the presence of the complementary oligonucleotide EY620 (Appendix A), as demonstrated by a 46% increase in fluorescence in the presence of Spike (Figure 3A). EY615QF unquenching after the addition of Spike indicates that EY615QF likely adopts an open structure when interacting with Spike.

Considering our LAMP results, which show Spike’s facilitation of the LAMP process (Figure 2), we hypothesize that this structural opening induced by Spike predominantly occurs at the 5′ end of the aptamer. This is because complete melting of the stem at the 3′ end would interfere with the LAMP process (Figure 1C), which relies on self-priming at the 3′ end. Therefore, if we take the energetically more probable structure model, the dumbbell-like structure (* in Figure 1A and upper configuration in Figure 3C), observations obtained with EY615QF are likely due to an opening of this structure at the 5′ end only (like depicted in Figure 3A), rather than at both the 5′ and 3′ ends simultaneously. An alternative explanation could involve the adaptation of other structures, such as structure #2 in Appendix A, where the two ends are far apart. This configuration could also account for the observed results, as it allows for self-priming at the 3′ end. Importantly, Spike binding may preferentially induce this structure over conformations where the extremities are in close proximity, similar to the opening effect described in the first scenario. This Spike-induced conformational change would explain both the EY615QF fluorescence increase and LAMP facilitation. This highlights the selective influence of Spike on the aptamer’s configuration, potentially favoring specific structures that enhance amplification efficiency.

#### 3.2.3. Spike Effect in Presence of Primers

We examined the influence of Spike on the EY615QF aptamer in the presence of LAMP primers EY616 and EY617 (Appendix A). With primers present, Spike induced a 31% increase in fluorescence (Figure 3B), comparable to the 46% increase observed without primers (Figure 3A). This finding is significant for two reasons. Firstly, it demonstrates that Spike affects fluorescence even when primers, present in excess relative to the aptamer, hybridize to the 3′ to open the predicted stem on this part of the aptamer EY615QF. This reinforces our earlier suggestion that Spike primarily opens the aptamer structure at the 5′ end, consistent with both the LAMP results and the EY615QF tests without primers (Figure 3A). The continued fluorescence enhancement by Spike, despite the 3′ end’s conformation being influenced by primers, strongly indicates Spike’s effect on the 5′ extremity. Secondly, Spike’s persistent effect in the presence of high primer concentrations (as used in LAMP and HRCA assays) confirms its continued interaction with EY615 under these conditions, which are compatible with IA assays.

#### 3.2.4. Interaction Model

Bringing together our experimental findings with the proposed interaction model of Song et al. [32], we put forth a plausible explanation for Spike’s role in modulating RCA and LAMP (Figure 3C). According to results summarized in Figure 3A, Spike seems to interfere with the “dumbbell-like” structure (i.e., it prevents the 3′ and 5′ extremities’ “head-to-head” posture). HRCA results support this observation, since Spike seems to inhibit RCA, which requires this 3′-5′ head-to-head structure to be initiated. This negative interference is likely due to the effect of Spike on either the 3′ and 5′ self-hybridization structures predicted for the aptamer (Figure 1A). In contrast, induction of LAMP by Spike (Figure 2) indicates that Spike seems to favor the 3′ self-priming structure, essential to the LAMP initiation (Figure 1C). All these results strengthen our hypothesis of an effect of Spike on the aptamer modifying the 5′ stem–loop while not interfering or even reinforcing the 3′ self-priming stem–loop structure (Figure 3C).

Our observations correlate with those of Song et al. [32] regarding molecular docking and molecular dynamics simulation (MDS)-based predictions about the aptamer’s potential interaction with Spike. According to this study, the nucleotides G3, C4, C28, T29, and G30, involved in 5′ self-hybridization, can interact with Spike (Figure 3C). Song et al. experimentally evaluated these computational predictions through point mutation studies. When nucleotides predicted to be critical for binding were altered, binding to Spike proteins was diminished. Interactions of these nucleotides with Spike might obstruct their participation in the base-pairing for folding of the 5′ stem, as pictured in Figure 1A, marked with an *, and Figure 3C. This information aligns with our mechanistic model and supports our observations of inhibited HRCA and increased fluorescence in the presence of Spike.

On the other hand, nucleotides T42 and T43, located nearer to the 3′ end of the aptamer sequence, are also predicted to interact with Spike [32]. However, these nucleotides would be in the loop of the 3′ stem and thus are not directly involved in the 3′ self-priming process according to predictions of secondary structures (Figure 1A and Figure 3C). As such, their interaction with Spike does not appear to be antagonistic to the formation of a 3′ self-priming structure or the unwinding necessary for LAMP. In fact, this interaction could plausibly facilitate these processes. In this context, we suggest that the interaction of Spike with these particular nucleotides might even favor the establishment of 3′ self-priming, which would explain the enhanced LAMP we observed in our experiments.

The hypothesized role of Spike, promoting the formation of some self-priming sites while hindering others, aligns with the unique requirements of RCA and LAMP. While RCA necessitates a head-to-head configuration of both the 5′ and 3′ ends, LAMP requires only 3′ self-priming. Our data suggest that Spike primarily affects the 5′ stem–loop, which could explain the inhibition of RCA via destabilization of the 5′–3′ interaction and the stimulation of LAMP production through opening of the 5′ end for primer elongation and stabilization of the 3′ end for self-priming. This model represents the most plausible interpretation of our collected data, offering a mechanistic understanding of how Spike differentially impacts these two processes. However, we acknowledge that while our evidence strongly supports this hypothesis, further structural studies would be necessary for absolute confirmation.

On a final note, the predictions of Song et al. [32] concerning the nucleotides involved in Spike–aptamer interaction and our proposed model for this interaction also correlate with another observation in LAMP tests. As proposed, when Spike interferes with the self-hybridization on the 5′ side of the aptamer, G30, which is the nucleotide matching the 3′ extremity of the LAMP primer (Figure 1C), would be rendered accessible (Figure 3C and Appendix A) and would thus help LAMP initiation. This suspected role of the C1–G30 base-pair interaction could also explain the results observed with LAMP assays using the EY615-5′C design; notably, the background signal (“-Spike” test results) is elevated and close to that of “+Spike” tests (Appendix A). With a design deprived of this C1 nucleotide (EY615-C5′), the 5′ stem of the aptamer would no longer affect the LAMP-initiating primer binding site (Appendix A). Thus, the role of Spike and its capacity to affect LAMP unwinding would become less critical, as we have observed in the LAMP tests conducted with the design EY615-5′C. Additionally, this deprivation of the cytosine in 5′ could affect the different structures adopted by the aptamer (Appendix A), reduce the probability of structures that compete with the one(s) that are LAMP-compatible, and thus decrease the dependency of the LAMP with regard to Spike presence. In any case, we should keep in mind that the structure proposed by Song et al. [32] is based on a predicted docking, which is itself based on a predicted structure of the aptamer and should not be considered equivalent to an experimentally determined structure; nevertheless, it does seem to be supported by our results.

Lastly, we should point out that our mechanistic studies were conducted at room temperature, which corresponds to the initial incubation steps of both RCA and LAMP. Although these conditions do not entirely mimic the high-temperature environment of the LAMP process, they do provide valuable insights into the Spike–aptamer interaction dynamics during the preliminary stages. Given the observed influence of Spike at room temperature, we infer that this effect might endure to a certain extent under the higher-temperature conditions typical of LAMP, although possibly at a reduced intensity. Most likely, initial amplification starts from the Spike-stabilized template, after which the amplification products themselves sustain the process. This is supported by our predictions of LAMP-compatible configurations at 60 °C (Figure 1), which could accumulate significantly once amplified with Spike’s initial assistance. This extrapolation allows us to bridge our room-temperature data with the higher-temperature LAMP context.

## 4. Discussion

Our model presents compelling evidence that an aptamer can trigger both IA systems, RCA and LAMP, in a ligand-dependent manner by using specifically tailored primers and leveraging the aptamer’s inherent ability to undergo conformational changes upon ligand binding.

We designed a model aptamer with a predicted dumbbell-like configuration at room temperature, featuring 5′ and 3′ extremities in a head-to-head position. This unique structure is suitable for initiating both RCA and LAMP processes. Our comprehensive analysis, including LAMP and RCA tests, configuration predictions, and structural studies using quencher-fluorescent conjugate aptamer tests, provided valuable insight into how this dumbbell-like structure is affected by Spike interaction. The results from these experiments converged to support a proposed mechanism for Spike-induced conformational changes in the aptamer, illuminating the potential of such an aptamer design to serve both LAMP and RCA allosterically. Through this model, we gained substantial evidence supporting the feasibility of using aptamers as dynamic triggers for isothermal amplification systems, opening new avenues for ligand-dependent nucleic acid amplification strategies.

In the context of LAMP particularly, this dumbbell-shaped design offers the additional advantage of bypassing the initial steps involving FIP and BIP primers in the classical LAMP process [3]. By utilizing this dumbbell-shaped aptamer, we directly initiate from an intermediate step of the traditional LAMP process (Figure 1), potentially streamlining the LAMP procedure.

Our experiments have demonstrated that this model allows both RCA and LAMP to occur in a Spike-dependent manner, but with opposing effects. Notably, we observed that the presence of Spike protein at a concentration of 50 nM inhibits RCA while simultaneously activating LAMP. This contrasting response to the same ligand highlights the versatility of our aptamer-based system.

To our knowledge, this study is the first to propose a model for a dynamic aptamer able to function as a LAMP template, in a ligand-dependent manner. The importance of using such dynamic aptamers for LAMP lies in the fact that LAMP presents several advantages over other amplification methods. LAMP demonstrates faster reaction times and higher amplification efficiency compared to both RCA and HRCA, typically achieving high-yield DNA amplification within 30–60 min [30]. This rapid amplification makes LAMP particularly suitable for time-sensitive diagnostic applications. Moreover, the increasing prevalence of LAMP-based point-of-care devices in the market suggests its superior efficiency under field conditions.

The study by Zhang et al. [31], albeit related to our study, follows a different trajectory, proposing an aptamer-based LAMP system where the aptamer is not used as the template initiating the LAMP process. They suggest an approach where an aptamer sets off a sequence of enzymatic events, generating primers essential for the LAMP system. This system, in the presence of the target ligand, continuously produces primers, making the ligand a true signal amplifier. Nonetheless, their approach uses split aptamers. Split-aptamer systems rely on two subunits, resulting in a more complex three-element kinetic system compared to our single-aptamer approach. This complexity can lead to less efficient interactions and potentially slower detection speeds [37]. Moreover, this configuration introduces certain challenges, particularly in terms of broadly adapting the concept to other aptamers [37,38] or to SELEX methods aimed at obtaining suitable aptamers for such methodology. Designing a system to select split aptamers is particularly challenging, as it requires the simultaneous optimization of two interacting fragments, adding another layer of complexity to the selection process [39].

Instead, we performed a very simple manipulation of the sequence to tune the dynamic conformation of an aptamer to craft a template for IA systems that operate in a ligand-dependent manner. Beyond the mere technical difference, this significantly increases its suitability for many potential applications and avenues for subsequent advancements. Most importantly, it paves the way for seamless integration with SELEX systems to generate compatible aptamers. Candidate aptamers that act as templates for RCA/LAMP can be easily selected and amplified in SELEX processes using these methods. This straightforward selection contrasts with the complexity of designing SELEX systems to select the cascading events required by approaches like that in Zhang et al.’s study [31].

Building on SELEX, our findings suggest that optimally designed aptamers (obtained, for example, with a specific SELEX, like in the case just mentioned) could operate on the same principles as our model aptamer but with enhanced ligand-dependent conformational variability for LAMP or RCA compatibility. Our model, while serving as a valid proof of concept, has limitations, as the ligand of interest (Spike) plays a supporting rather than critical role in triggering the conformational shifts leading to IA. Moreover, the current system requires the presence of 50 nm of Spike protein for detection, which is inadequate for clinical use, given that Spike concentrations in patient samples typically range from low to high picomolar. However, aptamers obtained through a specific SELEX selecting LAMP/RCA-triggering aptamers in a ligand-dependent manner would likely be more efficient and allow us to develop a highly sensitive system. By applying the lessons learned from our model to the SELEX process, we can design a selection strategy that specifically targets aptamers with enhanced ligand-dependent conformational changes, potentially leading to more sensitive and efficient LAMP/RCA-compatible aptamers for a wide range of other target ligands and diagnostic/detection applications. Our proof-of-concept study has provided valuable insights into designing such a SELEX system. We have learned that a library designed to adopt dumbbell-like shapes would be ideal for this purpose. This could be achieved by incorporating pre-determined nucleotide zones within the random sequences that allow for weak base-pairing at each extremity. Such a library design would enable the selection of aptamers where the ligand can either aid or disrupt the formation of the dumbbell-like structure, directly influencing the aptamer’s ability to trigger LAMP or RCA.

Such a SELEX process would not only enhance sensitivity but also provide the foundation for developing rapid diagnostic systems that capitalize on the dynamic nature of aptamers. Unlike the more common static use of aptamers in RCA and LAMP protocols, our approach of leveraging the dynamic features of aptamers offers significant advantages. Static methods often necessitate time-consuming procedures such as multiple washing steps, blocking, and extra incubation periods. By harnessing the dynamic nature of aptamers, the proposed method potentially saves 3–5 h in overall assay time, making it significantly faster and more efficient. This streamlined approach, combined with aptamers specifically selected for enhanced ligand-dependent conformational changes, could revolutionize the speed and sensitivity of aptamer-based diagnostics.

This investigation also highlights the feasibility of developing such systems from known aptamers using a rational design adaptation approach. Our findings set the stage for future research that promises sensitivity levels commensurate with the equilibrium dissociation constant (*K*_D_) of the aptamer, which commonly resides in the nanomolar range and can potentially extend to the pico- and femtomolar ranges [40]. Furthermore, the fact that we demonstrate that an aptamer can exhibit opposing responses in the presence of its target ligand, simultaneously enhancing LAMP while inhibiting RCA, may open the door to systems with dual-mode response. Given further development, these could, for example, lead to more robust diagnostic assays with built-in confirmation mechanisms, where the LAMP activation serves as the primary detection method and the RCA inhibition acts as a secondary confirmation. Overall, our study paves the way for the development of novel detection methods that leverage the unique properties of aptamers in isothermal amplification systems.

## Figures and Tables

**Figure 1 sensors-24-06875-f001:**
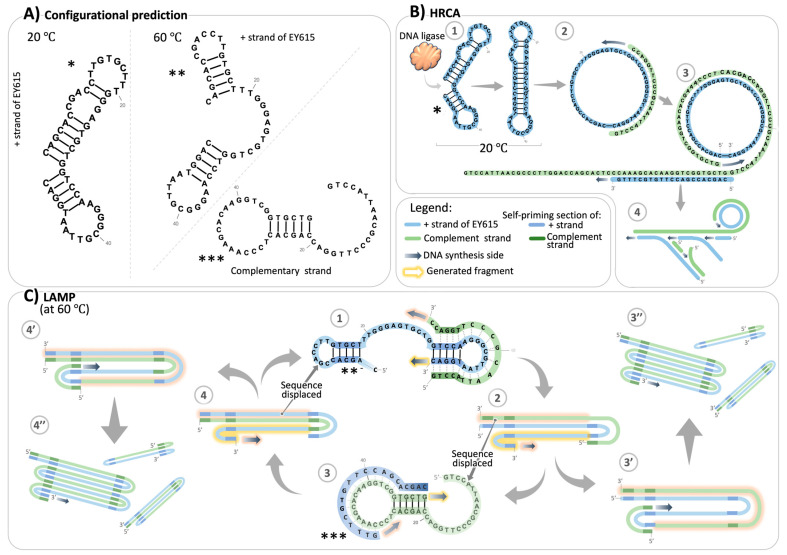
Structural predictions and isothermal amplification mechanisms of EY615 aptamer design. (**A**) Predicted structures compatible with requirements of the RCA or LAMP system. Asterisks denote structure predictions suitable for RCA (*), as well as LAMP (** for EY615 strand and *** for complementary strand). (**B**) HRCA schematic. A predicted structure can potentially initiate a ligation step, yielding circular DNA which can then initiate the RCA process. The primer pair used in our study can initiate a hyperbranched RCA, HRCA (step 4). (**C**) LAMP schematic. Structure prediction that can initiate a LAMP system. The resulting complementary reverse sequence of the Spike aptamer also presents a structure (step 3), to perpetuate LAMP. A portion of the resulting product of this DNA amplification loop contributes to products of different sizes (steps 2c and 4c).

**Figure 2 sensors-24-06875-f002:**
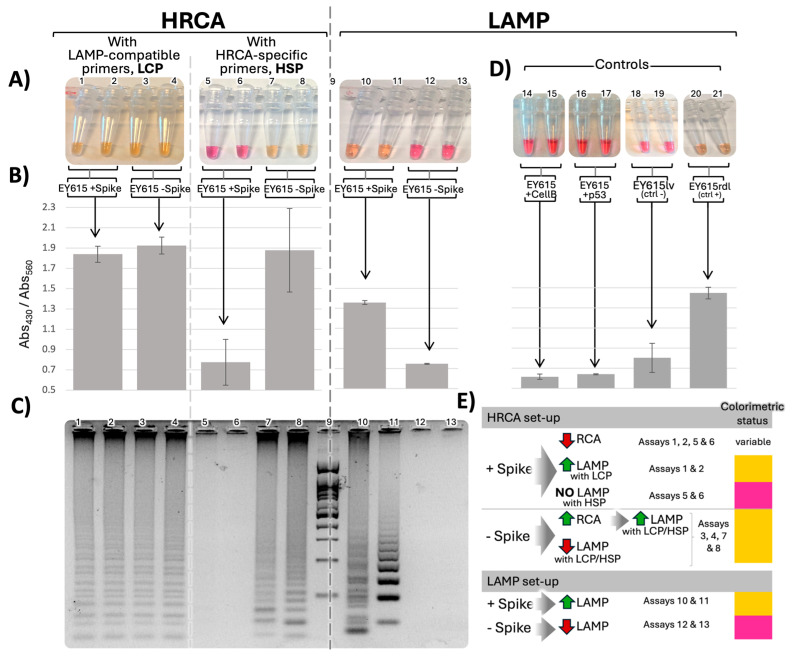
Spike-dependent HRCA and LAMP. (**A**) Colorimetric switch of EY615 aptamer duplicate assays in HRCA (40 min) or LAMP context (30 min) with or without Spike protein. (**B**) Average ratios of absorption intensities at 430 nm versus 560 nm for the IAs. (**C**) LAMP and HRCA product migration on 3% agarose gel. Lane 9: 100 bp ladder (Froggabio). (**D**) LAMP assays with EY615 incubated with control proteins, cellulase B (CellB) and p53, to assess if the effect observed with Spike is obtained with a random protein, negative control assay with the long-version aptamer (EY615lv) unable to form the dumbbell-like shape necessary for the HRCA and LAMP triggering, and positive control assay with a reinforced dumbbell-like structure forming sequence (EY615rdl). (**E**) Summary table of the different pathways leading to the observed outcomes in HRCA and LAMP processes. HRCA or LAMP activation is represented by green upward arrows, while inhibition is shown by red downward arrows. Gray arrows symbolize the consequential effect between events. The table shows also the assays corresponding to the combination or succession of events depicted and the last column shows the expected colorimetric results.

**Figure 3 sensors-24-06875-f003:**
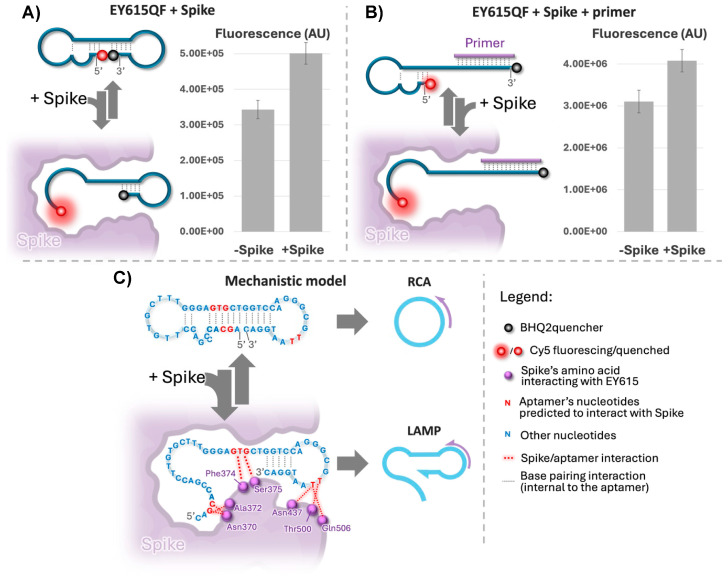
Mechanism of interaction of the EY615QF aptamer design with Spike. (**A**) Fluorescence of EY615QF aptamer (6.5 nM) with and without Spike protein (100 nM). (**A**,**B**) Left section: anticipated fluorescence events; right section: fluorescence measurements. (**B**) Fluorescence of EY615QF interaction with RCA/LAMP primers (EY616 and EY617 at 3µM each; sequence details in Appendix A) in the presence/absence of Spike protein (100 nM). (**C**) Theoretical structure model of the EY615QF aptamer and Spike protein interaction and its hypothetical impacts on RCA and LAMP initiations. Representation of nucleotides and amino acids as in Song et al. [32].

## Data Availability

The original contributions presented in this study are included in the article; further inquiries can be directed to the corresponding author.

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
