# Peer review of "Adaptation of a Model Spike Aptamer for Isothermal Amplification-Based Sensing"

_sensors, 2024, doi:10.3390/s24216875_

Round 1

Reviewer 1 Report

Comments and Suggestions for Authors

Thank you very much for the opportunity to review the manuscript titled "Adaptation of a Model Spike Aptamer for Isothermal Amplification-Based Sensing". In this paper, the authors developed a novel isothermal amplification-based sensing system by adapting an aptamer targeting the SARS-CoV-2 Spike protein. They exploited aptamer conformational changes to trigger amplification processes like LAMP and HRCA, establishing a ligand-responsive detection method for broader molecular targets. The subject area of the manuscript is quite interesting, and it would certainly add a scientific contribution to the relevant field. I recommend the publication in "Sensors" after minor revision. The following suggestions are provided for the authors' revising manuscript.

1.      The abstract lacks details on key performance indicators. What is the sensitivity, accuracy, and speed of your aptamer-triggered LAMP and RCA systems compared to standard diagnostic methods?

2.      The authors claim that their system is simpler and faster than existing IA aptamer-triggered assays. Could the authors provide more detailed comparisons, particularly in terms of reaction times, ease of implementation, and the system's robustness under field conditions?

3.      The manuscript needs a more thorough language polishing, and it may require professional editing exactly in vocabulary collocation and the connection between the sentences is not very smooth specially in the introduction section.

4.      Please ensure that the labeling (A, B, C) across all figure panels is consistent in size and formatting. Currently, the sizes of these labels are not uniform, which may impact the clarity of the figures.

5.      I am not satisfied with the conclusions of this manuscript, please rewrite it to improve the quality of your work.

6.      Some references are too old! For example, 1995, 2000, 2002 and so on, please update the whole manuscript with latest literature. 

Comments on the Quality of English Language

Moderate editing of English language required.

Reviewer 2 Report

Comments and Suggestions for Authors

The article presents an aptamer-based system that triggers isothermal amplification (LAMP and RCA) in response to Spike protein, with structural changes observed primarily at the 5′ end. This novel approach offers ligand-dependent activation and could streamline diagnostic assays. The study highlights aptamer design optimizations, potential dual-mode detection, and implications for broader diagnostic applications using SELEX-selected aptamers. Authors should consider the comments below while revising the manuscript.

1.          Could the aptamer systems be fine-tuned to detect other ligands beyond Spike, and what modifications would be necessary for broader diagnostics?

2.          Were the predicted Spike-aptamer binding sites experimentally validated, for example, through point mutations in the interacting nucleotides?

3.          Have control experiments with modified aptamers been conducted to test whether Spike selectively opens the 5′ stem-loop while preserving the 3′ structure?

4.          How does Spike's influence on the aptamer change at elevated LAMP temperatures, and are there predictions of shifts in binding affinity or folding?

5.          What limits the current aptamer from being more efficient in triggering LAMP, and could truncations or other modifications improve its sensitivity?

6.          Could excess primers cause steric hindrance that limits the Spike-aptamer interaction, and what is the optimal primer-to-aptamer ratio?

7.          Have other Spike-targeting aptamers been tested, and are there ones with stronger binding affinities or more efficient conformational changes?

8.          What specific steps in SELEX optimization are targeted, and would high-throughput techniques or machine learning play a role in improving aptamer selection?

9.          How does this single-aptamer system compare with other aptamer-based methods like split aptamers in terms of sensitivity and ease of use?

10.      Could tuning the aptamer’s secondary structure with modified nucleotides enhance its ability to undergo ligand-dependent conformational changes?

11.      What are the anticipated limitations when using this system in clinical or point-of-care settings, and how can those challenges be mitigated?

Round 2

Reviewer 2 Report

Comments and Suggestions for Authors

I recommend acceptance of the revised manuscript for publication.